# Development of Nanocomposite Materials Based on Conductive Polymers for Using in Glucose Biosensor

**DOI:** 10.3390/polym14081543

**Published:** 2022-04-11

**Authors:** Lyubov S. Kuznetsova, Vyacheslav A. Arlyapov, Olga A. Kamanina, Elizaveta A. Lantsova, Sergey E. Tarasov, Anatoly N. Reshetilov

**Affiliations:** 1Laboratory of Biologically Active Compounds and Biocomposites, Tula State University, Lenin pr. 92, 300012 Tula, Russia; l.s.latunina@gmail.com (L.S.K.); o.a.kamanina@gmail.com (O.A.K.); dart.liza@yandex.ru (E.A.L.); 2Institute of Biochemistry and Physiology of Microorganisms of the Russian Academy of Sciences, Federal Research Center “Pushchino Scientific Center for Biological Research of the Russian Academy of Sciences”, Pushchino, pr. Science, 5, 142290 Moscow, Russia; setar25@gmail.com (S.E.T.); anatol@ibpm.pushchino.ru (A.N.R.)

**Keywords:** conductive polymers, electron transfer mediators, polyaniline, poly(thionine), poly(neutral red), single-walled carbon nanotubes, thermally expanded graphite, glucose oxidase

## Abstract

Electropolymerized neutral red, thionine, and aniline were used as part of hybrid nanocomposite conductive polymers, to create an amperometric reagent-less biosensor for glucose determination. The structure of the obtained polymers was studied using infrared (IR) spectroscopy and scanning electron microscopy. Electrochemical characteristics were studied by cyclic voltammetry and impedance spectroscopy. It was shown that, from the point of view of both the rate of electron transfer to the electrode, and the rate of interaction with the active center of glucose oxidase (GOx), the most promising is a new nanocomposite based on poly(neutral red) (pNR) and thermally expanded graphite (TEG). The sensor based on the created nanocomposite material is characterized by a sensitivity of 1000 ± 200 nA × dm^3^/mmol; the lower limit of the determined glucose concentrations is 0.006 mmol/L. The glucose biosensor based on this nanocomposite was characterized by a high correlation (R^2^ = 0.9828) with the results of determining the glucose content in human blood using the standard method. Statistical analysis did not reveal any deviations of the results obtained using this biosensor and the reference method. Therefore, the developed biosensor can be used as an alternative to the standard analysis method and as a prototype for creating sensitive and accurate glucometers, as well as biosensors to assess other metabolites.

## 1. Introduction

At present, biosensor technologies are rapidly developing and play an important role in the field of clinical analysis, pharmaceuticals, the food industry, and environmental monitoring [1,2,3,4]. Enzyme sensors are among the most popular biosensors and are widely used in various fields, due to their high specificity and sensitivity in complex systems. In particular, biosensors for determining glucose have been widely used in the field of clinical diagnostics for a long time and are familiar, convenient portable devices, which have almost completely replaced the alternative methods of analysis [5]. The main direction of recent biosensor research has been the optimization of the process of electron transfer between the enzyme and the electrode surface, and the improvement of the characteristics of biosensors [6,7]. Significant efforts are being made to find new electrode surface modifiers or enzyme immobilization matrices, as a good modifier or immobilization matrix can prevent leakage and preserve enzyme activity, which will lead to improved biosensor capabilities [8].

Currently, nanocomposites using conductive polymers and nanomaterials are interesting and promising materials for modifying the surface of electrodes [9,10]. Conductive polymers are widely used in biosensors of both catalytic and affine type. An extensive list of works on DNA sensors using conjugated conductive polymers is presented in the review [11]. In the field of electrochemical sensors, conductive polymers can be used as an immobilizing matrix for the biological material, and can also serve as an electron transport conductor. Conductive polymers make it possible to create reagent-free biosensors, they are resistant to biodegradation, and at the same time have biocompatibility, are easy to use, and, therefore, are attractive for use in the design of biosensors [12]. The properties of conductive polymers can be finely adjusted using special methods of organic synthesis [9]. It is common knowledge that metals have a high electrical conductivity and organic substances are insulators; however, electrically conductive polymers combine the properties of both, also having the advantage of better processability. In conducting polymers, π-electrons are delocalized along the entire main chain, and the transfer of electrons between microorganisms and the electrode is carried out along the entire conductive polymer framework. Electrically conductive polymers usually outperform classical mediator systems in terms of electron transfer rate, since during their operation there is no need for physical movement of a charged particle between the electrode and the biomaterial, and, accordingly, there is no leaching of electroactive compounds from the biocatalytic layer of the sensor [1].

Another important factor that should be taken into account when developing biosensors based on conductive polymers should be mentioned. Often, the polymer is weakly attached to the surface of the working electrodes (most often based on graphite), which significantly reduces the efficiency of electron transport, which occurs due to the kinetic limitation of the surface electron transfer to the electrode [13]. Nanomaterials can increase the surface area for the formation of the conductive polymer and enhance its attachment to the electrode. In the field of biosensor analysis, the use of carbon nanotubes (CNTs) and other nanomaterials makes it possible to achieve a high rate of electrocatalytic processes [14,15,16].

Polyaniline (PANI), poly(neutral red) pNR, and poly(thionine) (pTN) can be distinguished among the polymers used to create the electrochemical systems of biosensor sensors and biofuel cells. Structures based on PANI are quite widely used in the design of biosensors for glucose determination, due to their useful properties, such as resistance to environmental influences, adjustable conductivity, large surface area, and redox properties [17]. For example, a composite based on PANI-montmorillonite with the inclusion of platinum nanoparticles was used to create a highly selective and stable glucose biosensor [18]. The biosensor demonstrated excellent long-term stability over two months and was successfully applied to human serum glucose measurements. The linear range of glucose determination was 10 μM–1.94 mm. PANI can also effectively work in hybrid nanocomposites; for example, with polydiphenylamine [19]. The synergistic interaction of the components of the hybrid nanocomposite provides enhanced electron transfer, to carry out the process of glucose oxidation. It should be noted that the conductivity of PANI is strongly affected by the pH of the solution; it shows higher conductivity in acidic solutions, and lower conductivity in a neutral solution [20].

Non-toxic semiconductor films made of pNR provide high catalytic currents in enzymatic reactions [21]. Glucose oxidase (GOx) exhibits excellent catalytic activity in the presence of glucose, upon the addition of nanocomposites based on pNR [22]. Thus, a nanocomposite bioanode based on pNR, graphene, and gold nanoparticles was created in article [23]. The modified bioanode showed a fast and sensitive response to changes in glucose concentration in the linear range of 10–100 mM. Developments in the creation of biosensors for determining glucose based on pTN have been less frequent. Thus, Ghika et al. successfully carried out the electrochemical polymerization of pTN on carbon film electrodes modified with CNTs and formed a biosensor for the determination of glucose and uric acid [24]. The biosensor for determining glucose allowed measurements in the linear range of 0.2–1.2 mM. Moreover, pTN has been successfully used in the design of biosensors for the determination of other analytes, such as clinically-significant lung cancer markers [25], nicotinadenindinucleotide (NADH) [26], ascorbic acid and uric acid [27], and acetaminophen [28].

Despite the large number of works published on this topic, the search for optimal solutions for the formation of sensitive, selective, and cheap glucose biosensor sensors is still ongoing [6]. In addition, the analysis of blood glucose is to some extent a classic model experiment, which can compare new approaches to the formation of receptor systems, which can then be used in a wide range of catalytic biosensors to assess other indicators. In previous works, we showed the possibility of successfully using nanomaterials as part of biosensors and their positive effect on the characteristics of electrodes [16,22,29]. In this work, we studied the effect of nanomaterials on the electrochemical characteristics and parameters of an amperometric glucose biosensor, using the example of CNTs and thermally expanded graphite (TEG) in nanocomposite matrices containing electrochemically polymerized pNR, pTN, and PANI. A more complete understanding of the mechanism of action of nanocomposite materials under the conditions of a functioning biosensor systems is possible with a kinetic study of the electron transfer process, with a study of the impedance characteristics of electrodes modified with nanomaterials, as well as with the use of an electron microscopic method for studying the surface topology.

## 2. Materials and Methods

### 2.1. Materials

We used the enzyme GOx (Sigma-Aldrich, St. Louis, MI, USA) (specific activity 15 U/mL). Neutral red (Dia-m, Moscow, Russia), thionine (Dia-m, Moscow, Russia), and aniline (Dia-m, Moscow, Russia) were used as the basis of the conducting matrix. Conducting nanocomposite systems were formed and biosensor measurements were performed using a sodium-potassium phosphate buffer solution with a pH of 6.8 (33 mM KH_2_PO_4_ and 33 mM Na_2_HPO_4_, Dia-m, Moscow, Russia).

### 2.2. Synthesis of Conducting Polymers

#### 2.2.1. Electrochemical Polymerization of Aniline on the Surface of the Working Electrode

Aniline was electrochemically polymerized by cycling the applied potential from −0.1 to + 0.9 V relative to the reference electrode (Ag/AgCl) at a scanning rate of 20 mV/s for 10 cycles in a solution containing 0.1 M aniline in 0.1 M HCl [30]. The structure of the resulting conductive polymer is shown in Figure 1a.

#### 2.2.2. Electrochemical Polymerization of Neutral Red on the Surface of the Working Electrode

Initially, the electrode was subjected to a preliminary electrochemical treatment by cycling the applied potential from 0.0 to +1.0 V relative to the reference electrode (Ag/AgCl) in 0.1 M KNO_3_ solution for at least 10 cycles, until stable cyclic voltammograms were obtained. Neutral red was polymerized electrochemically by cycling the applied potential from −1.0 to +1.0 V relative to the reference electrode at a scan rate of 50 mV/s for 20 cycles in 0.05 M sodium potassium phosphate buffer solution (pH = 5.6), containing 1 mM neutral red and 0.1 M KNO_3_ [32]. The structure of the resulting conductive polymer is shown in Figure 1b.

#### 2.2.3. Electrochemical Polymerization of Thionine on the Surface of the Working Electrode

First, the electrode was subjected to preliminary electrochemical treatment by cycling the applied potential from 0.0 to +1.0 V relative to the reference electrode (Ag/AgCl) in 0.1 M KNO_3_ solution for at least 10 cycles, until stable cyclic voltammograms were obtained. Thionine was electrochemically polymerized by cycling the applied potential from −1.0 to +1.0 V relative to the reference electrode at a scan rate of 50 mV/s for 30 cycles in a solution containing 0.025M NaB_4_O_7_ (pH = 9.0), 1mM thionine, and 0.1 M KNO_3_ [24]. The structure of the resulting conductive polymer is shown in Figure 1c.

### 2.3. Formation of Nanocomposite Materials Based on the Obtained Conductive Polymers

Electrodes modified with CNTs were obtained by applying 1 µL of a suspension of single-walled CNTs (the length of CNTs was 1–10 µm, the average diameter was 1.5 nm, the external specific surface was 450 m^2^/g, OOO Uglerod Chg, Russia) on surface of the working electrode and left to dry completely. A 0.1-mm-thick layer of TEG (synthesized according to the method in [33]) was formed on a working graphite electrode with a diameter of 3 mm by pressing at 150 bar. The resulting TEG had a bulk density of 16 g/L. Electrochemical polymerization of the conforming conductive polymers was carried out on the surface of graphite printed electrodes modified with TEG and CNTs. After the synthesis of conducting polymers, the electrodes were washed with a sodium-potassium phosphate buffer solution (pH = 6.8) and dried.

### 2.4. Formation of Working Electrodes Based on Conductive Polymers and Nanocomposite Materials Based on Them

GOx (3 μL) enzyme was applied to the dried surface of electrodes modified with conductive polymers or their nanocomposites and left to dry completely. A mixture was prepared from 0.0035 g of bovine serum albumin (BSA), 50 µL of sodium-potassium phosphate buffer solution (pH = 6.8), and 7.5 µL of glutaraldehyde. This mixture (3 µL) was applied to the surface of the electrodes and left to dry until completely dry.

### 2.5. Electrochemical Measurements

An EmStat potentiostat (PalmSens, The Netherlands) was used for electrochemical measurements. We used graphite printed electrodes (GPE) (OOO Rusens, Russia) made according to a three-electrode scheme (Figure 2).

Measurements were carried out in potassium-sodium-phosphate buffer solution with pH 6.8 at 22 °C. The cell volume was 4 mL. Stirring was carried out with a magnetic stirrer (ZAO Ekros, Russia) at a speed of 200 rpm. Cyclic voltammograms were recorded at a potential sweep rate of 20–200 mV/s. The measured parameter (response of the biosensor) in the amperometry mode was the amplitude of the output signal of the biosensor after the substrate was added. After each measurement, the electrode was washed three times with a buffer solution.

### 2.6. Impedance Spectroscopy

Electrochemical measurements were carried out in a 2 mL cuvette at a temperature of 25 °C; and with constant stirring (500 rpm). The background solution was 25 mM potassium phosphate buffer pH 6.5 containing 10 mM sodium chloride. The impedance characteristics were recorded using a VersaSTAT 4 galvanopotentiostat impedance meter (Ametek Inc., Berwyn, PA, USA). The impedance characteristics were measured at an applied potential of +30 mV (vs. Ag/AgCl) in the frequency range from 40 kHz to 0.2 Hz with a voltage modulation amplitude of 10 mV. A suitable equivalent electrical circuit for each system was selected using the ZSimpWin program (EChem Software, Warminster, PA, USA).

### 2.7. IR Spectroscopy

An FMS 1201 infrared Fourier spectrometer (OOO Monitoring, Russia) was used to obtain IR spectra. The spectra were recorded in a KBr tablet (OOO Dia-m, Moscow, Russia) in the region of 4000–500 cm^−1^ at a ratio of the mass of the conductive matrix sample to the mass of potassium bromide 2:300 (mg).

### 2.8. Scanning Electron Microscopy (SEM)

A target-oriented approach was utilized for optimization of the analytic measurements [34]. Before measurement, the samples were mounted on a 25-mm aluminum specimen stub and fixed by graphite adhesive tape. Metal coating with a thin film (10 nm) of gold/palladium alloy (60/40) was performed using a magnetron sputtering method, as described earlier [35]. The observations were carried out using a Hitachi SU8000 field-emission scanning electron microscope (FE-SEM). Images were acquired in secondary electron mode at a 5 kV accelerating voltage and at a working distance 8–10 mm. The morphology of the samples was studied taking into account the possible influence of metal coating on the surface [35].

## 3. Results

### 3.1. Formation of Nanocomposite Materials Based on Conductive Polymers

Aniline, neutral red, and thionine were used as monomers to obtain conductive polymers. These redox compounds form stable conducting polymers upon electropolymerization [24] and are characterized by nontoxicity and biocompatibility with living systems [9,36]. Single-walled CNTs and TEG were used to form nanocomposite materials. CNTs are one of the most commonly used types of CNTs, due to their high conductivity [37]. Modification of the surface of graphite printed electrodes with a wide range of carbon nanomaterials (single-walled and multi-walled CNTs with different functionalizations) with different parameters can significantly increase the sensitivity of the sensors being created [7]. TEG, as a nanomaterial, has a large surface area compared to conventional graphite, which allows it to be used in biosensors to increase sensitivity and operating range. The advantages of TEG are its low cost and manufacturability compared to other carbon-based fillers, such as CNTs and nanofibers [38]. Previously, a number of articles described the use of TEG in the development of biosensors for the determination of ethanol [39], ascorbic acid [40], and propionaldehyde [41].

The principal scheme of operation of such a system consists in the reaction of the conducting matrix with the reduced enzyme, followed by electron transfer to the electrode surface (Figure 2). At the same time, the nanomaterial significantly increases the area of contact between GOx and the conductive polymer, increases the overall conductivity of the system, and contributes to the good retention of GOx on the surface.

The structures of PANI, pNR, and pTN obtained by electrochemical polymerization were studied using IR spectroscopy (Figure 3).

The resulting PANI matrix was characterized by a peak at 3440 cm^−1^, which corresponds to asymmetric stretching of N-H bonds; an absorption band at 2900 cm^−1^ corresponds to stretching of unsaturated aromatic C–H bonds; and peaks in the region of 2300 cm^−1^ correspond to stretching of C_sp_^3^-H bonds in the benzoid group. The peak at 1605 cm^−1^ indicates the protonated emeraldine form and corresponds to C=C quinoid vibrations. The peak at 1499 cm^−1^ corresponds to C=C benzoid vibrations; at 1306 cm^−1^, absorption of the C–N bond; at 1248 cm^−1^, vibrations of aromatic C–H bonds; at 1128 cm^−1^, stretching vibrations of C=NH^+^; and at 822 cm^−1^, to out-of-plane bending vibrations of C–H bonds of benzoid rings. The resulting bands of the initial PANI matrix confirm its polymerization on the electrode surface [42].

The resulting pNR polymer can be identified by the presence of an absorption band at 3440 cm^−1^, which corresponds to the stretching vibrations of the N-H bonds in the phenylenamine structures (C–NH–C). The band at about 2900 cm^–1^ in the polymer refers to asymmetric stretching vibrations of CH_3_ groups [23]. The presence of the band at 1380 cm^−1^ corresponds to vibrations of C_sp_^3^–H bonds [43]. The band at 2850 cm^−1^ characterizes the stretching vibrations of the –N(CH_3_)_2_ bonds. The intensity of this band is extremely low, which is due to the fact that one methyl group is cleaved from the –N(CH_3_)_2_ group during oxidative polymerization. The intense band at 1630 cm^−1^ corresponds to C–C stretching vibrations of aromatic rings. The presence of an absorption band in the region of 940 cm^−1^, as well as at 1080 cm^−1^ (out-of-plane bending vibrations of C–H bonds of the 1,2,4,5-substituted benzene ring) indicates that the growth of the polymer chain is achieved by adding C–N between 3-amino groups and the para-position of the phenyl rings with respect to nitrogen, as in the polymerization of aniline, which occurs according to the ‘head’ to ‘tail’ type [44]. After analyzing the IR spectrum, we can say that the IR spectrum corresponds to pNR. The peak at 2924 cm^−1^ in the IR spectrum of pTN refers to C–H stretching vibrations of aromatic ring hydrogen atoms. The peak at 1643 cm^−1^ is associated with in-plane deformation of –NH_2_, due to the binding of the nitrogen atom of the primary amino group to the phenothiazine ring. The broad band at 3449 cm^−1^ is consistent with the presence of amino groups, due to the presence of thionine. Weak absorption bands in the region 1400–1300 cm^−1^ refer to C–N bonds. The bending in the C–H plane is evidenced by weak absorption bands in the region of 1000 cm^−1^ [45]. IR proved that the polymerization of thionine had passed, a polymer had formed on the electrode, but some quantity of thionine was still present, possibly due to its high adsorption on graphite. The IR data correspond to the works in [46,47].

The structure of the studied conducting hydrogels was studied using scanning electron microscopy. The resulting images are shown in Figure 4.

The used printed electrode has a highly developed surface (Figure 4a), which provides a large area of contact between the enzyme and the conductive matrix and makes it possible to achieve a high sensitivity of the biosensors based on the developed electrodes. When using CNTs and TEG, they are evenly distributed on the electrode surface, which greatly facilitates the transfer of electrons from the enzyme to the electrode (Figure 4b,c). The structure of TEG is similar to sheets of natural exfoliated graphene [48]. Graphite nanosheets are thin flakes of graphite layers with smooth and uniform surfaces. The synthesized polymers were successfully electropolymerized on the electrode surface, the three-dimensional structure of pNR and PANI films with porous morphology is clearly visible, as can be seen from Figure 4d,f. The electrode surface is practically invisible under the layer of electropolymerized polymers, which is explained by the growth of the polymer chain directly on the graphite materials, and is consistent with the results of previously described studies [42,49].

### 3.2. Electrochemical Properties of Electrodes Modified by Created Nanocomposites

The study of the electrochemical properties of graphite electrodes modified with conductive polymers and their nanocomposites is important for understanding the mechanism of their signal generation and the principles of their interaction with biological materials. Better knowledge of the mechanisms of electron transport between biological materials and modified polymers will allow the development of more efficient and easy-to-use biosensor analyzers, which will be characterized by a long service life and a high stability of analysis results. Chemically conducting polymers are organic cyclic polymer compounds with a pronounced semiconductor electronic structure [50]. In the electrochemical system under study in the absence of nanomaterials, two stages can be distinguished: electron transfer along the polymer molecule due to delocalized electrons, and electron transfer from the conducting polymer to the electrode (Figure 2). The introduction of nanomaterials into the structure of conductive polymers increases the conductivity of polymers and the contact area of the conductive matrix with the electrode surface and biomaterial; that is, the speed of both components of the electrochemical process increases.

The electrochemical properties of the resulting conductive gels were studied using cyclic voltammetry (CV). This method is convenient for finding limiting stages of electrochemical processes, since the limiting anode current is directly proportional to the square root of the sweep rate in the case of slow electron transfer through a conducting nanocomposite. If this stage occurs fairly quickly and the process is limited by the surface reaction on the electrode, the current is proportional to the sweep rate [51]. Typical types of studied cyclic dependences for the example of pTN are shown in Figure 5.

During polymerization on the electrode of the pTN mediator, redox peaks appear on the CV corresponding to the polymerization product (Figure 5a) [52]. For electrodes modified with CNTs and pTN, an increase in the anodic peak is observed in comparison with an electrode based on dissolved thionine, which also indicates an improvement in the conductivity of the system in the presence of nanomaterials and the facilitation of electron transfer to the electrode. The identification of the limiting stage made it possible to apply the Nicholson model (Equation (1)) [51] and the Laviron model [53] (Equation (2)) to find the heterogeneous rate constants of electron transfer to the electrode (Table 1). The use of the Nicholson and Laviron models in determining the electrochemical properties of conductive and redox-active polymers can be found in [54,55], respectively: (1)$$k_{S}=\psi \sqrt{\pi \frac{nFv}{RT}D},$$
(2)$$\log(k_{S})=\alpha \log(1-\alpha )+(1-\alpha )\log\alpha -\log(\frac{RT}{nFv})-\frac{\alpha (1-\alpha )nF\Delta E}{2.3RT},$$
where *k_s_* is the heterogeneous rate constant of the electrochemical system (s^−1^ cm); ψ is the parameter affecting the potential difference of the peaks (∆*E*, mV); *n* is the number of participating electrons; *F* is the Faraday number (C/mol); *ν* is the potential sweep rate (V/s); *R* is the universal gas constant (J·mol/K); *T* is temperature (K); *D* is the diffusion coefficient (cm^2^/s); π is constant 3.14; α is the transfer coefficient of the cathode process; (1 − α) is the transfer coefficient of the anode process; ∆*E* is the potential difference between the anode and cathode peaks (V). 

In addition, for the practical use of conducting nanocomposite systems, it is important to study, not only the electrochemical aspects of electron transfer to the electrode, but also the features of the interaction of conducting polymers with the enzyme used. The interaction constants of the enzyme with the conducting polymers were found using the Nicholson–Shine simulation, to study the possibility of using the created nanocomposites for GOx immobilization [56]. This model is widely used to analyze the rate constant of the interaction of mediators, including those in conducting polymers, and with various enzymes [16,57,58,59]. This model considers the change in the limiting anode current before and after the introduction of an oxidizable substrate into the system. The rate of the biochemical stage of interaction of the mediator with the biomaterial has a pseudo-first order, with an excess of substrate concentration, and the interaction constant is related to the change in the limiting anodic current by the Nicholson and Shine equation (Equation (3)):(3)$$\frac{I_{k}}{I_{d}}=\sqrt{\frac{k_{\mathrm{int}}\left[E\right]RT}{nFv}},$$
where *I_k_* is the limiting current in the presence of the substrate (A), *I_d_* is the limiting current in the absence of the substrate (A), *k_int_* is the rate constant of the interaction between the mediator and the biomaterial (dm^3^/(mg s)); *R* is the universal gas constant, J/(mol K); *T* is temperature (K); [*E*] is the enzyme titer (mg/L); ν is sweep speed (V/s); *n* is the number of transferred electrons; and *F* is the Faraday constant (C/mol). A typical view of the obtained curves is shown in Figure 5b, and the values of the calculated constants are presented in Table 1.

Thus, nanocomposite matrices based on TEG conjugated with pNR and pTN have the highest heterogeneous electron transfer rate constants. pNR and pTN are sufficiently efficient electron carriers [59], and TEG significantly increases the contact area of the conducting polymer with the electrode. In addition, since the technology for obtaining TEG includes the stage of treating graphite with a mixture of concentrated nitric and sulfuric acids, its surface may contain carboxyl groups [60,61], which can form hydrogen bonds with the amino groups of pNR and pTN, promoting closer contact and better electron transfer in the nanocomposite. The low rate constants of interaction with GOx for matrices based on PANI are due to a decrease in its conductivity at neutral pH [20], which is optimal for GOx.

From the point of view of the interaction of the obtained nanocomposite matrices with GOx, it can be noted that the introduction of a nanomaterial, practically does not change the interaction constant of the nanocomposite with GOx, which confirms the proposed electron transfer model based on the interaction of the GOx active center with the conducting matrix (Figure 6). The interaction constants of GOx with the obtained conducting nanocomposites are close to the constants of the interaction of GOx with soluble mediators [57,62], which indicates the high efficiency of the electron exchange between the synthesized polymers and flavinadenindinucleotide (FAD) in the active center of GOx. Based on the values of the obtained interaction constants of the conductive polymer with the biomaterial, it can be seen that the system based on pNR has the highest rate constant. The high value of the interaction rate constant in this case is associated with the effectiveness of NR in systems with GOx [23,63]. Thus, based on the obtained kinetic constants, the most promising are electrodes based on pNR-TEG and pTN-TEG nanocomposites.

The conductivity of the measuring electrode is the most important parameter of an electrochemical biosensor, because it directly affects the rate of charge transfer from the active centers of the enzyme of microorganisms to the electrode in the process of transformation of the substrate. Each component of the biosensor contributes to this process. In the described variant of biosensor, these components are CNTs or TEG, which modify the electrode surface, an enzyme, a conductive polymer, and BSA, which hold the biocatalyst on the electrode surface. We used the method of electrochemical impedance spectroscopy to assess the effect of each of the components on the overall conductivity of the system. Each of the components was immobilized on the surface of a graphite electrode, separately and in various combinations with other components, and the impedance spectra of the resulting compositions were studied. Figure 6a shows the impedance spectra in the form of Nyquist plots (−Zim vs. Zre) for various variants of electrode modification, using the example of a composite of TEG and pNR. Figure 6b shows an enlarged high-frequency region for the same variants of electrode modification. A standard Randles equivalent circuit (inset) was used to interpret the resulting impedance spectra. It consists of charge transfer resistance (Rct), ohmic resistance of the solution (Rs), and surface double-layer capacitance (Cdl). The high frequency semicircle on the Nyquist plot is ascribed to the electron-transfer limited process, and its diameter depicts the Rct.

It can be seen from the obtained diagrams that the addition of nanomaterial (TEG) and pNR to the electrode surface sharply reduced the total resistance of the electrode. The electrode with the pNR-TEG nanocomposite demonstrated the lowest total resistance; the value of the charge transfer resistance was 245 Ohm. The increase in resistance upon the introduction of GOx is associated with the low resistance of BSA, which is used for immobilization of the enzyme. At the same time, it should be noted that, when glucose was added to the measuring cuvette, the resistance of the electrode with the pNR-TEG and GOx nanocomposite decreased from 352 Ohm to 255 Ohm, which indicates the presence of electron transfer in the system when the substrate was introduced. Similar results were also obtained for other nanocomposites.

### 3.3. Practical Use of Biosensors Based on Obtained Nanocomposites

Enzyme-based biosensors are catalytic-type receptors; the response in such systems is provided by enzymatic reactions. Analytical and metrological characteristics were obtained by constructing calibration dependences of biosensor responses on glucose concentration at a constant applied potential. A typical form of dependence for nanocomposites based on pNR is shown in Figure 7.

The dependences of the sensor response on the glucose concentration have a hyperboidal form and were approximated using the Michaelis-Menten equation:(4)$$V=\frac{V_{\max}{\left[S\right]}^{}}{K_{M}{}^{}+[S]},$$
where V_max_ is the maximum rate of the enzymatic reaction, at which all enzyme molecules participate in the formation of the enzyme–substrate complex, and is achieved at [S]→∞; K_M_ is the effective Michaelis constant, numerically equal to the substrate concentration at which the rate of the enzymatic reaction reaches half of the maximum value (V = Vmax/2); V is the reaction rate; [S] is substrate concentration, mol/dm^3^.

It follows from the Michaelis-Menten Equation (4) that the analytical signal is proportional to the glucose concentration at low substrate concentrations, which makes it possible to single out a linear section of the calibration curve that is limited from above by the K_M_ value. The results for determining the linear section of the calibration dependence are presented in Table 2.

The electrode obtained during this work, modified with a nanocomposite based on pNR and TEG, has the highest sensitivity. It should be noted that the developed biosensors are characterized by high sensitivity to glucose. This electrode surpasses most of the described analogues in terms of the value of the sensitivity coefficient and the lower limit of the determined concentrations [18,22,23,29,63], which shows the promise of the approach proposed in this work for the creation of nanocomposites based on conductive polymers and carbon nanomaterials.

It is important to note that the values of the lower limits of the determined glucose concentrations generally correlate with the results of determining the rate constants of the interaction of the mediator with GOx and the electrode (Table 1). At the same time, the heterogeneous rate constant of electron transfer to the electrode has a stronger effect on the characteristics of the biosensor, which is probably due to the fact that, in the developed system, it limits the process of interaction of the conducting matrices with the surface of the working electrode.

In addition, the sensors retained a response value of at least 75% of the initial level for more than 7 days. However, this is not a limiting parameter for disposable sensors for monitoring blood glucose.

Comparative tests were carried out to determine the content of glucose in human blood using a biosensor based on a conducting nanocomposite pNR-TEG system. A certified OneTouchSelect glucometer (LifeSkan Inc. (Johnson and Johnson), Milpitas, CA, USA) was used as a reference method. Blood samples were taken at different times, before and after meals, and the results are shown in Figure 8. Statistical processing (modified Student’s test) of the results showed that the data obtained by both methods differed slightly. Thus, the developed biosensor can be used as an alternative to the standard analysis method and as a prototype for creating sensitive and accurate glucometers.

## 4. Conclusions

In the work, a number of conductive gels were synthesized, including carbon nanomaterials. These gels were used to create reagent-less glucose biosensors. The obtained graphs of the cyclic current-voltage dependence during the polymerization of neutral red mediator, thionine, and aniline confirmed the presence of these fragments in the composition of the polymerization product and their decisive contribution to the reactivity of the biosensor in redox interactions. A biosensor with a matrix based on a nanocomposite of pNR and TEG was identified as the most promising and suitable for further use as part of reagent-less mediator biosensors, in accordance with their superior kinetic constants and metrological characteristics. This biosensor outperforms the other biosensors obtained in this work and literary analogues, for such parameters as the lower limit of detectable concentrations (C_LOD_ = 0.006 mmol/dm^3^) and detection limit (C_min_ = 0.002 mmol/dm^3^). Statistical analysis did not reveal any deviations of the results obtained using the biosensor and the reference method. Therefore, the developed biosensor can be used in clinical studies for fast and accurate measurement of blood glucose levels. In addition, it is important to note that the methodological approach presented in this paper for the creation of nanocomposites based on conductive polymers can be easily applied to the formation of other electrochemical biosensors characterized by a high sensitivity and accuracy of analysis.

## Figures and Tables

**Figure 1 polymers-14-01543-f001:**
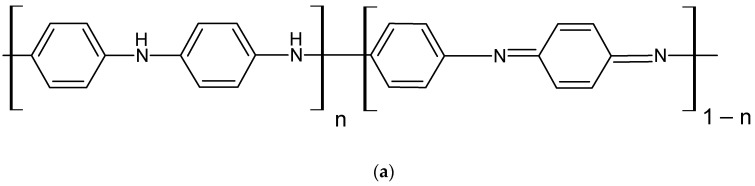
Structure of (**a**) polyaniline (PANI) [31], (**b**) poly(neutral red) (pNR), (**c**) poly(thionine) (pTN) obtained by electrochemical polymerization.

**Figure 2 polymers-14-01543-f002:**
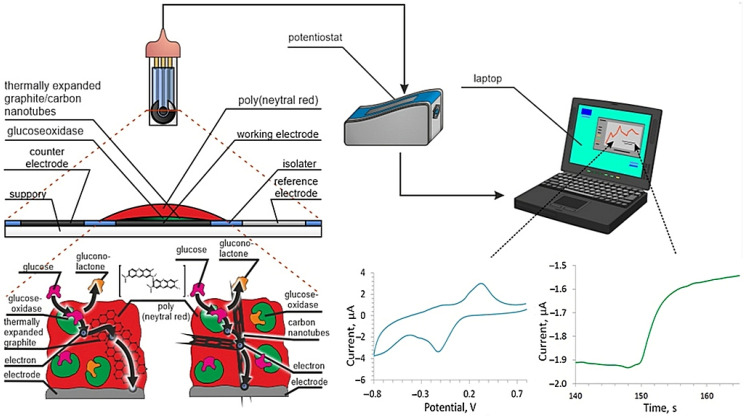
Scheme of the used three-electrode graphite printed electrode and biosensor setup for glucose determination and structure of electrodes modified with nanocomposite materials based on a conductive polymer poly (neutral red).

**Figure 3 polymers-14-01543-f003:**
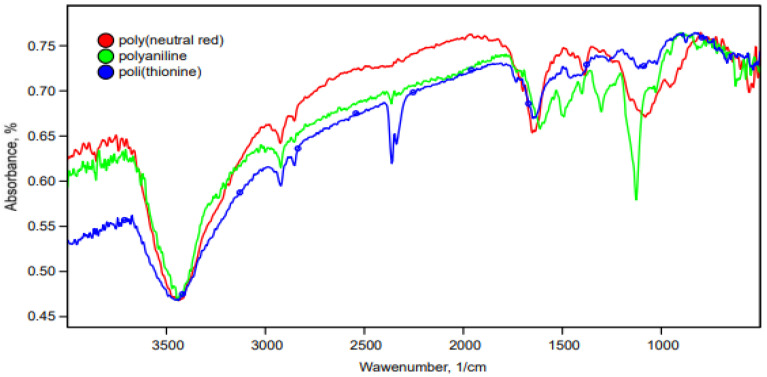
IR spectra of the resulting conductive polymers.

**Figure 4 polymers-14-01543-f004:**
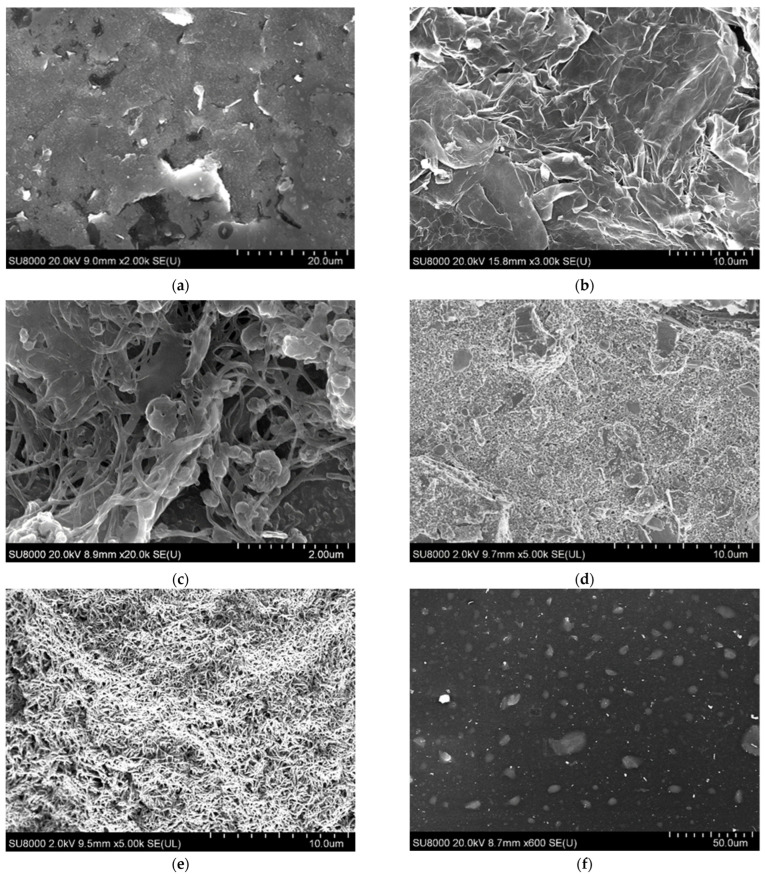
Scanning electron microscopy for electrodes modified with conductive polymers and nanocomposite materials: (**a**) clean printed electrode; (**b**) electrode modified with TEG; (**c**) electrode modified with CNTs; (**d**) electrode modified with pNR; (**e**) electrode modified with PANI; (**f**) electrode modified with CNTs, pNR, and covered with a bovine serum albumine (BSA) membrane.

**Figure 5 polymers-14-01543-f005:**
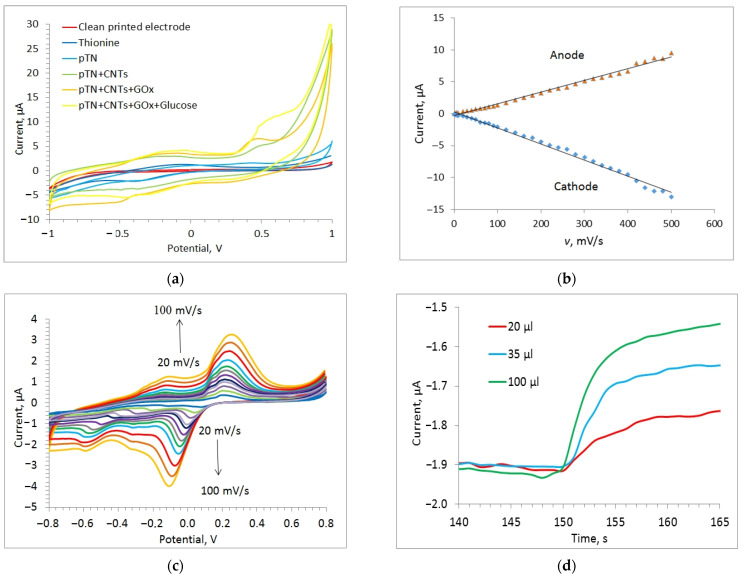
Electrochemical studies of the printed electrode with the developed conducting nanocomposite systems based on thionine: (**a**) cyclic voltammogram (CVA) at different stages of electrode modification; (**b**) dependence for determining the transfer coefficients (α) in p(TN)-CNTs; (**c**) CVA for the nanocomposite p(TN)-CNT system at different potential scan rates; (**d**) typical biosensor signal of a biosensor with a nanocomposite p(TN)-CNT material with the addition of different concentrations of glucose.

**Figure 6 polymers-14-01543-f006:**
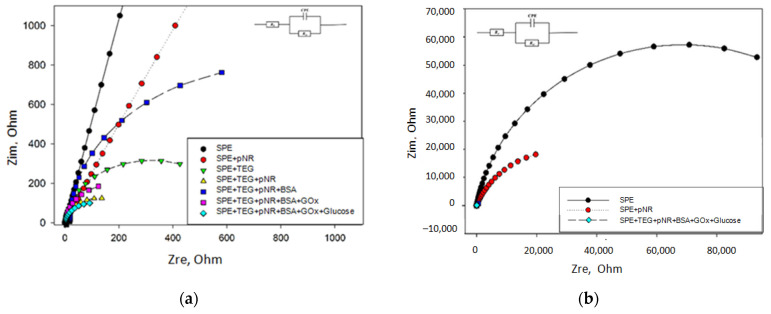
Nyquist plots from EIS results for various composites (**a**). Increased high frequency region for Nyquist diagrams (**b**).

**Figure 7 polymers-14-01543-f007:**
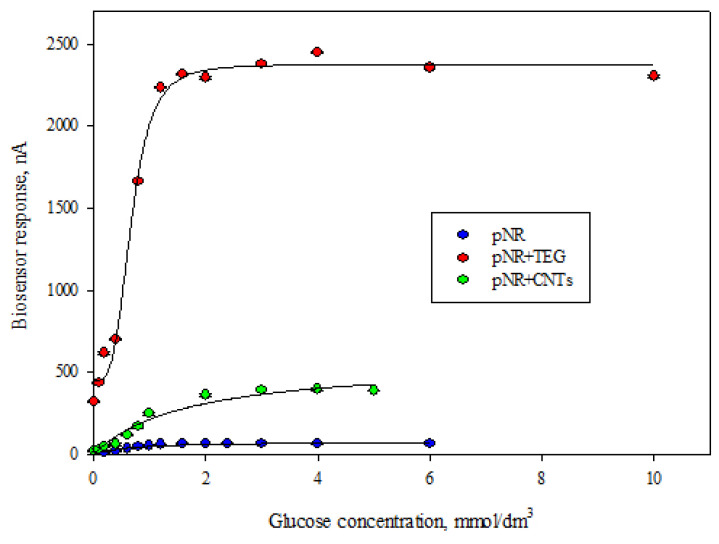
Calibration dependences of biosensors based on pNR and its nanocomposites.

**Figure 8 polymers-14-01543-f008:**
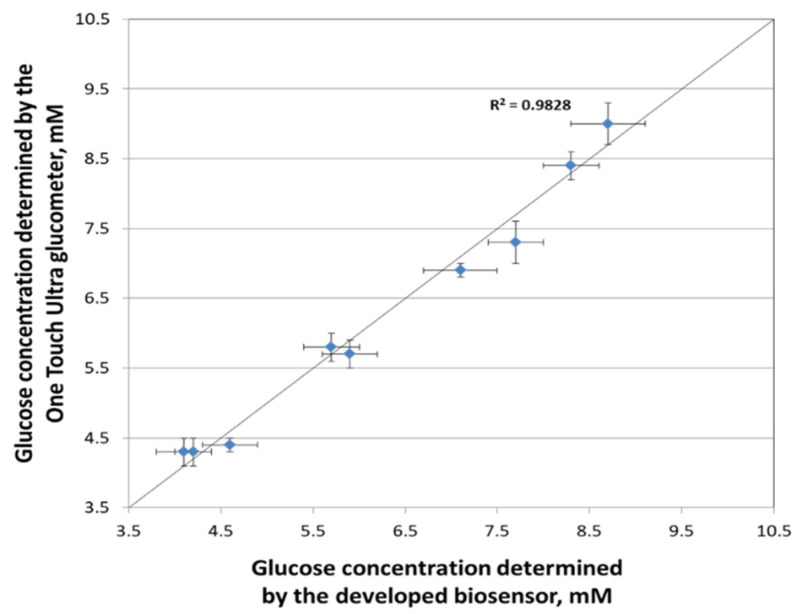
Determination of blood glucose content using biosensors based on the created conducting systems.

**Table 1 polymers-14-01543-t001:** Heterogeneous rate constants of electron transfer and rate constants of interaction with GOx for the developed conducting polymers and their nanocomposites.

Matrix	Operating Potential, mV	Heterogeneous Rate Constant, (s^−1^·cm)	Rate Constant of Matrix Interaction with GOx, cm^3^/(mol·s)
pNR	−550	0.35 ± 0.02	14 ± 2
pNR-TEG	−550	1.43 ± 0.03	13 ± 3
pNR-CNTs	−550	0.90 ± 0.05	11 ± 3
pTN	350	0.90 ± 0.02	5.9 ± 0.3
pTN-TEG	350	1.61 ± 0.08	6.2 ± 0.3
pTN-CNTs	350	0.14 ± 0.01	6.1 ± 0.3
PANI	450	0.46 ± 0.06	0.1 ± 0.01
PANI-TEG	450	0.57 ± 0.09	0.1 ± 0.01
PANI-CNTs	450	0.55 ± 0.07	0.1 ± 0.01

**Table 2 polymers-14-01543-t002:** Main characteristics of biosensors based on the developed modified printed electrodes and analogues.

Conductive Matrix	Range of Determined Concentrations, mmol/dm^3^	Sensitivity Coefficient nA × dm^3^/mmol	Detection Limit (C_min_) mmol/dm^3^	Time of a Single Measurement, min
pNR	0.09–1.0	62 ± 4	0.03	1–2
pNR-TEG	**0.006–0.5**	**1000 ± 200**	0.002	1–2
pNR-CNT	0.02–4.2	220 ± 10	0.007	1–2
pTN	0.25–1.9	165 ± 3	0.08	1
pTN-TEG	0.06–0.62	450 ± 20	0.02	1
pTN-CNT	0.2–1.7	125 ± 2	0.07	1
PANI	0.5–0.72	11 ± 1	0.15	1–2
PANI-TEG	0.4–1.4	23 ± 5	0.12	1–2
PANI-CNT	1.5–2.1	17 ± 2	0.23	1–2
BSA covalently linked to ferrocene and containing CNT-NH_2_ [63]	0.1–1.8	330 ± 10	–	1–3
A conjugate of reduced graphene oxide and Fe_3_O_4_ nanoparticles [64]	0.05–1.0	5900	–	–
Conducting hydrogel based on organic-inorganic hybrid sol-gel matrix and CNT [29]	0.045–1.04	1480 ± 30	–	1–3
Platinum nanoparticles incorporated into PANI-montmorillonite hybrid composites [18]	0.01–1.94	–	0.0001	–

## Data Availability

Not applicable.

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
