# Peer review of "Development of Nanocomposite Materials Based on Conductive Polymers for Using in Glucose Biosensor"

_polymers, 2022, doi:10.3390/polym14081543_

Round 1
Reviewer 1 Report
The work presented in this manuscript is novel, interesting, and well-suited for publication in polymers. It reports on the use of a new nanocomposite based on poly(neutral red) (pNR) and thermally expanded 20 graphite (TEG) as a new biosensor for detection of glucose. There are just a few points which need to be addressed before this publication can be accepted.
- The introduction is lacking in review of DNA biosensors based on different detection methods besides graphene and CNTs. For example, a popular detection technique for DNA biosensors uses cationic conjugated polymer to form complexes with DNA to produce a colorimetric or fluorescence response. It would be important to compare with different detection methods in the introduction as well. Please expand on the introduction, and here is a review you can consider citing (http://dx.doi.org/10.1016/B978-0-12-803581-8.10144-4).
- The term TEG is used frequently in this article however it is not a commonly known term and is not properly explained in the introduction. The title of the manuscript is too lengthy and difficult to find for readers in the field. Please try to shorten.
- I suggest combining figures 1, 2, and 3 into 1 figure for better overview of chemical structures of all conjugated polymers used in this paper. The structure does not need to be this big for one chemical.
- Please also combine figures 4 and 5. The readers are interested to know how the detailed structure of the electrode together with an overview of the entire setup for a more complete understanding.
- In figure 10, please explain the fairly high (~450nA) biosensor response for pNR+TEG biosensors even without any glucose.

Reviewer 2 Report
The author present a paper about the possible development of a biosensor of glucose. The introduction as well as the experimental part are well written, the authors explain well the materials and the setups giving to the reader the whole vision. The discussion part is complete and well organized.
My main concerned is relatively to the term "bio". I think the authors abuse of this term without doing any bio test. If a polymer is bio does not impose that the application has to be biocompatible. So I suggest to take out, mostly in the title, the bio word "Based on Conductive Biocompatible Polymers for Use in Biosensor..."; in the conclusions or perspective you can state that if the CP is biocompatible so also the final device <might> be.
Secondly the authors have to revise the figures, mostly the chemical structures: Figure 1 is the only one in between 1,2,3 which is showing a polymer, please look in literature and paste a correct chemical structure, respecting the bond angles. In Figure 2 the left ring of the right tricycle is badly connected to the amine.
In Figure 4 the zoom of the graph out of the laptop is useless: if you'd like to report please make it visible at leas the scales.
In Figure 6 intensity in% in a FTIR is weird, either absorbance or transmittance.
Reviewer 3 Report
In this work, reagent-free glucose biosensors have been prepared, based on the electrical response of conductive gels based on hybrid nanocomposites of conductive polymers, and the desired objectives have been achieved, especially with the poly(neutral red)-based biosensor, ( pNR) and thermally expanded graphite (TEG), which surpassed other biosensors in key parameters (such as the detection limit), both prepared in this work and in previously published works, according to the data presented by the authors. The characterization of the response of this biosensor enables it to be used in clinical studies for the rapid and precise measurement of blood glucose levels, which leaves no room for doubt regarding the technological interest of the article. In addition, as a study of the interaction of conductive polymer nanocomposites with the enzymatic biological system is also presented, the work also provides a well-founded methodological approach of great scientific interest.
The presentation of the work seems correct to me and is well structured. The preparation methodology of the conductive polymer nanocomposites and thermally expanded graphite is described in detail, as well as the metrological kinetic characterization of the prepared sensors. The instruments used in the structural and micromorphological characterization, as well as the electrical response and the analytical parameters, are of good technological level, and all the analytical methodologies followed have been clearly exposed.
As for the images, they are in a reasonable order of appearance, they are clear, and the size of the numbers and letters of the axes and the legends of the graphs seems sufficient to me. As for the quality of the English writing of the manuscript, I do not consider myself qualified to assess it. I always recommend, as a general rule, a thorough spelling and syntax check. I have not found any noteworthy formal error.
The approach and definitions that make up the introduction of the work have a correct order of presentation and are duly referenced.
Finally, it should be noted that the conclusions of the work are correct and are in line with the objectives set out in the introduction, having achieved, as I have already mentioned, a successful result.
Only one doubt appears to me, within the framework of the important technical application of the prepared sensor, in the field of clinical analysis, and it is the following: for the preparation of the biosensor, the glucose oxidase enzyme is fixed in the pNR matrix, obtaining a very useful configuration for blood glucose analysis. Now, can pNR cause some enzyme degradation effect over time? In other words, what is the stability and/or durability of the biosensor obtained with respect to that of other conventional sensors? I consider important the incorporation to the manuscript of some temporary stability test, if it is available, or alternatively, a theoretical reflection regarding the temporary stability of the biosensor based on the structure and composition of the prepared nanocomposite gel.
The authors must develop this point, in the terms that I have described, so that, in my opinion, the work is suitable for publication in this journal.
Round 2
Reviewer 1 Report
The comments have been sufficiently addressed and the manuscript is ready to be accepted in its present form.